# Hierarchical Masked Autoregressive Models with Low-Resolution Token Pivots

Guangting Zheng [1][†]   Yehao Li [2]   Yingwei Pan [2]   Jiajun Deng [3]   Ting Yao [2]   Yanyong Zhang [1]   Tao Mei [2]

## Abstract

Autoregressive models have emerged as a powerful generative paradigm for visual generation. The current de-facto standard of next token prediction commonly operates over a single-scale sequence of dense image tokens, and is incapable of utilizing global context especially for early tokens prediction. In this paper, we introduce a new autoregressive design to model a hierarchy from a few low-resolution image tokens to the typical dense image tokens, and delve into a thorough hierarchical dependency across multi-scale image tokens. Technically, we present a Hierarchical Masked Autoregressive models (Hi-MAR) that pivot on low-resolution image tokens to trigger hierarchical autoregressive modeling in a multi-phase manner. Hi-MAR learns to predict a few image tokens in low resolution, functioning as intermediary pivots to reflect global structure, in the first phase. Such pivots act as the additional guidance to strengthen the next autoregressive modeling phase by shaping global structural awareness of typical dense image tokens. A new Diffusion Transformer head is further devised to amplify the global context among all tokens for mask token prediction. Extensive evaluations on both class-conditional and text-to-image generation tasks demonstrate that Hi-MAR outperforms typical AR baselines, while requiring fewer computational costs. Code is available at https://github.com/HiDream-ai/himar.

[†] This work was performed when Guangting Zheng was visiting HiDream.ai as a research intern. [1]University of Science and Technology of China, Anhui, China [2]HiDream.ai Inc, Beijing, China [3]The University of Adelaide, Adelaide, Australia. Correspondence to: Yingwei Pan <panyw.ustc@gmail.com>, Yanyong Zhang <yanyongz@ustc.edu.cn>.

*Proceedings of the 42nd International Conference on Machine Learning*, Vancouver, Canada. PMLR 267, 2025. Copyright 2025 by the author(s).

## 1. Introduction

In recent years, GPT-style Autoregressive (AR) models have brought a powerful revolution in Natural Language Processing (NLP), and become the model of choice in designing Large Language Models (LLMs) (Achiam et al., 2023; Team et al., 2023; Bai et al., 2023; Dubey et al., 2024; Team et al., 2024). The dominant training paradigm in AR models is to predict the next word/token in a sequence conditioned on previous estimated words, i.e., next token prediction. This prevailing paradigm has shown excellent scalability via scaling laws and generalization in zero-shot settings, paving a reliable way towards Artificial General Intelligence (AGI).

Inspired by the scaling successes of AR models in NLP, a steady of attempts have been attained to scale up AR models in CV field for visual generation (e.g., text-to-image generation (Sun et al., 2024; Tian et al., 2024; Li et al., 2024; He et al., 2025; Lu et al., 2024a) and text-to-video generation (Ren et al., 2024b; Wang et al., 2024b;a)). Specifically, after decomposing input image into a sequence of image patches/tokens, one direction (Esser et al., 2021; Lee et al., 2022; Sun et al., 2024; Zhuo et al., 2024) directly adopts GPT-style AR models with causal attention (see Figure 1 (a)), which enforces next token prediction by only attending to preceding tokens. Another direction commonly adopts BERT-style AR models with bidirectional attention (see Figure 1 (b)), e.g., Masked Autoregressive model (MAR) (Li et al., 2024; Fan et al., 2024), that simultaneously predicts multiple masked tokens in a random order by attending to all masked and unmasked tokens. These AR models with next token prediction objective, while being dominant in NLP, have not yet been scaled effectively in visual generation, especially for visual content creation with complex semantics. Accordingly, in large-scale text-to-image/video generation, typical AR models tend to be less effective when compared to other generative models (e.g., Diffusion models (Rombach et al., 2022; Dhariwal & Nichol, 2021; Esser et al., 2024; Zhang et al., 2024; Chen et al., 2023) and Diffusion Transformer (Peebles & Xie, 2023; Bao et al., 2023; Lu et al., 2024b; Zhu et al., 2024)). This might be attributed to two inherent limitations in typical AR models: (1) Most AR models capitalize on vector quantization process to decompose images into discrete tokens, thereby inevitably resulting in information loss. (2) Both GPT-style and BERT-style AR models solely hinge on a single-scale sequence of

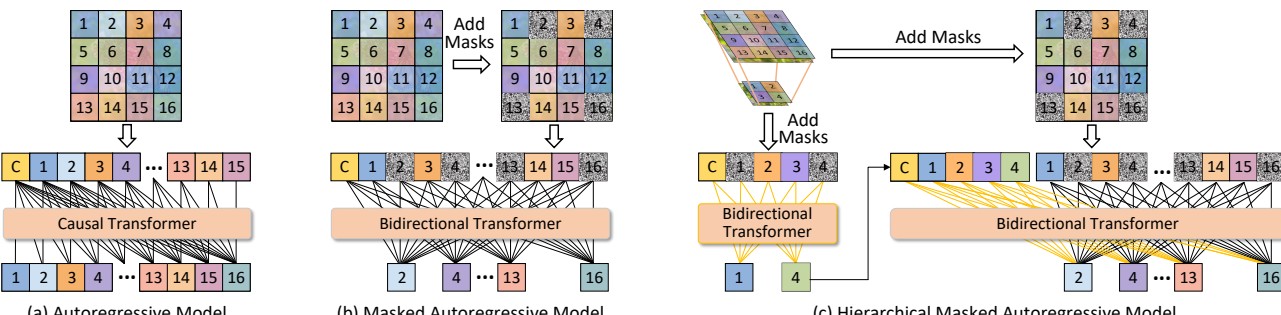

*Figure 1.* **a) Next-token autoregressive (AR).** GPT-style autoregressive models process 2D image tokens as a 1D sequence, predicting tokens in raster order using causal attention to ensure each token depends only on preceding ones. **b) Next-token masked autoregressive model (MAR).** BERT-style autoregressive models initially consider all tokens to be masked, subsequently predicting each masked token based on the known tokens in a random order, leveraging bidirectional attention to enable parallel prediction of a subset of tokens. **c) Hierarchical mask autoregressive model (Hi-MAR).** Hierarchical mask autoregressive models adapt a hierarchical prediction strategy to address the lack of global context in the next-token prediction. Hi-MAR first predicts a low-resolution image token sequence, which contains a few tokens, to reflect the global structure, and then pivots on these tokens to enhance and refine the next-resolution prediction.

image tokens for predicting next tokens in autoregressive form. Such single-shot autoregressive design makes it difficult to capture global context for facilitating early token prediction, resulting in sub-optimal autoregressive modeling among all dense image tokens.

In an effort to mitigate the above limitations, we start from the recent pioneering work of MAR that nicely sidesteps vector quantization and triggers autoregressive modeling in a continuous-valued space. This lossless image tokenization seems more suitable for autoregressive modeling in vision data. It motivates us to delve into the potential of global context mining in MAR for boosting visual generation, thereby further alleviating the second limitation. In order to upgrade MAR with additional global context, we derive a particular form for autoregressive modeling named Hierarchical Masked Autoregressive models (Hi-MAR). Our launching point is to build a top-down hierarchical structure from the root of low-resolution image tokens to the leaf layer of typical dense image tokens. Figure 1 (c) conceptualizes such construction of hierarchical structure in Hi-MAR, which enables hierarchical autoregressive modeling in a multi-phase fashion. Specifically, the first phase performs bidirectional autoregressive modeling over low-resolution image tokens. Such learnt low-resolution image tokens naturally reflect global structural information of the whole image. After that, we take the low-resolution image tokens as intermediary pivots to guide the next autoregressive modeling over typical dense image tokens. In this way, Hi-MAR could elegantly trigger global context propagation in a top-down manner, and the second-phase autoregressive modeling over typical dense image tokens does benefit from this additional guidance of global context. Meanwhile, this additional global context propagation significantly eases the autoregressive modeling over dense image tokens, and thus requires fewer generation steps, thereby harboring an innate agency that remains advantageous in the speed of sequence prediction.

Moreover, in the second phase, we design a new Diffusion Transformer head to mine global context among all masked and unmasked tokens, aiming to further strengthen autoregressive modeling over dense image tokens.

The main contribution of this work is the mining of global context in masked autoregressive models for image generation. The solution also leads to the elegant view of how to build and interpret the global structure of an image, and how to nicely integrate such global structural awareness into typical masked autoregressive models, which are problems not yet fully understood in the literature. Through extensive experiments on ImageNet and MS-COCO, we demonstrate the effectiveness of our proposal, for example, achieving 0.38 FID performance boost over MAR and only requiring 54% computational costs.

## 2. Related Works

**Diffusion models.** In the domain of image generation, diffusion models (Ho et al., 2020; Rombach et al., 2022; Peebles & Xie, 2023; Esser et al., 2024; Qian et al., 2024; Wan et al., 2024) have emerged as a powerful paradigm, achieving impressive results across diverse tasks. These models conceptualize image generation as a denoising process (Ho et al., 2020), where images are progressively reconstructed from noise through a multi-step denoising strategy. A common backbone for diffusion models is the CNN-based U-Net (Song & Ermon, 2019; 2020), which effectively captures both local and global features during denoising. To improve scalability and flexibility, DiT (Peebles & Xie, 2023) replaces the U-Net backbone with a transformer (Waswani et al., 2017). Building on this, U-ViT (Bao et al., 2023) incorporates long skip connections inspired by U-Net into the transformer architecture, and treats all inputs as tokens to enhance performance.

**Next-token autoregressive models.** Autoregressive models (AR) have achieved significant success in natural language processing (Achiam et al., 2023; Team et al., 2024; Bai et al., 2023; Dubey et al., 2024; Brown et al., 2020; Radford et al., 2019) and are increasingly applied to image generation (Zhuo et al., 2024; Team, 2024; Sun et al., 2024; Tian et al., 2024; Chang et al., 2022). Early works, such as VQVAE (Esser et al., 2021) and RQ-Transformer (Lee et al., 2022), serialize 2D images into 1D token sequences and predict tokens in raster order. LlamaGen (Sun et al., 2024) adapts this next-token prediction paradigm and scales up models to achieve quality comparable to diffusion models. Due to the fact that the dependencies between 2D image tokens do not naturally follow a raster order, some studies (Chang et al., 2022; Yu et al., 2023) suggest that causal attention may not be ideal for image generation. To address this, models like MaskGIT (Chang et al., 2022), MagViT (Yu et al., 2023), MagViT-2 (Yu et al., 2024), and MUSE (Chang et al., 2023) adopt bidirectional attention and masked prediction strategies used in BERT (Kenton & Toutanova, 2019), enabling non-sequential token prediction.

Compared to mainstream continuous-valued diffusion models, traditional autoregressive models, inherited from language autoregressive models, predict over discrete tokens, which may introduce information loss, ultimately limiting the generation quality. Recently, GIVT (Tschannen et al., 2025) and MAR (Li et al., 2024) replaced discrete tokens with continuous tokens. MAR introduces a diffusion loss function to model per-token probabilities, replacing the categorical cross-entropy loss used in discrete-valued models. Furthermore, Fluid (Fan et al., 2024) extends continuous-valued autoregressive models to text-to-image generation.

**Next-scale autoregressive models.** Multi-scale autoregressive generation has been explored in various forms to enhance image synthesis quality (Chang et al., 2023; Tian et al., 2024; Tang et al., 2025; Ren et al., 2024a). Among them, Muse (Chang et al., 2023) generates high-resolution discrete tokens conditioned on low-resolution tokens and text inputs, while VAR (Tian et al., 2024) replaces next-token prediction with a next-scale prediction strategy by retraining a multi-scale quantization autoencoder to encode an image into K discrete token maps at different resolutions. However, such approaches often rely on VQ-based quantization, which can introduce substantial information loss and limit the fidelity of generated outputs (Li et al., 2024). Moreover, retraining multi-scale autoencoders, as in VAR, increases model complexity and training costs. In contrast, our approach generates continuous-valued token sequences from multiple resolutions of images, eliminating the need for additional autoencoder training and improving the upper bound of generation quality.

Another limitation of existing approaches lies in how they model cross-scale dependencies. VAR (Tian et al., 2024) and HART (Tang et al., 2025) employ a shared Transformer to autoregressively model tokens across scales without explicit scale-specific guidance, which may underutilize the unique characteristics of each resolution level. Muse (Chang et al., 2023) adopts two separately trained models for low-resolution and high-resolution generation, increasing parameter count. FlowAR (Ren et al., 2024a) introduces spatially adaptive layer normalization in the flow matching head, facilitating position-by-position semantic injection and enabling scale-wise adjustments; however, it lacks an explicit mechanism for encoding scale-level information within the Transformer backbone. In contrast, our method mitigates this by introducing an elegant scale-aware Transformer backbone that incorporates explicit resolution information while maintaining parameter efficiency. To explicitly encode scale information, we introduce a learnable scale vector for each resolution, which is injected into the Transformer backbone via adaLN-Zero operations (Peebles & Xie, 2023), further enhancing scale-awareness. This explicit scale encoding allows the model to better capture resolution-specific characteristics and improves multi-scale generation quality.

In addition to the information loss caused by token discretization and the challenges in modeling cross-scale dependencies, a critical limitation shared by existing multi-scale models (Chang et al., 2023; Tian et al., 2024; Tang et al., 2025; Ren et al., 2024a) is the reliance on ground-truth low-resolution tokens during training to supervise high-resolution generation. This introduces a discrepancy between training and inference since ground-truth tokens are not available at inference time. Our approach mitigates this issue by conditioning high-resolution generation on predicted low-resolution tokens from the Transformer backbone, ensuring consistency between training and inference and improving generation robustness. Furthermore, methods like HART adopt an MLP-based diffusion head that treats each token independently, leading to a loss of global information during the denoising process. In contrast, our method introduces a Diffusion Transformer head that leverages self-attention to capture inter-token dependencies during diffusion, resulting in more coherent generation.

## 3. Method

In this work, we devise Hierarchical Masked Autoregressive models (Hi-MAR) that pivot on low-resolution image tokens to trigger hierarchical autoregressive modeling in a multi-phase manner. This section starts with the preliminaries of MAR (Li et al., 2024), which triggers autoregressive modeling in a continuous-valued space. Then, the hierarchical masked autoregressive model is elaborated. After that, we detail the integration of a newly proposed Diffusion Transformer head, which mines the global context among

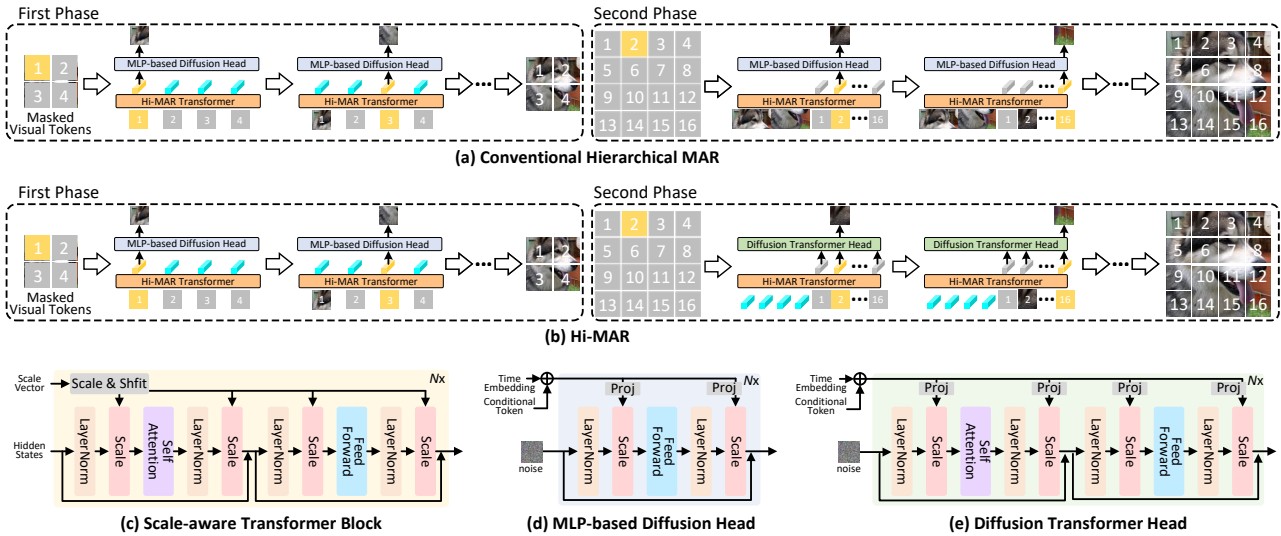

*Figure 2.* **(a) Pipeline of conventional hierarchical MAR.** Conventional hierarchical mar uses a shared Transformer for both phases and directly leverages low-resolution visual tokens to guide second-phase predictions. **(b) Pipeline of Hi-MAR.** An image and its low-resolution counterpart are converted into token sequences at two scales. During inference, both sequences are initially masked. In the first phase, masked low-resolution tokens are processed by the Transformer to predict conditional tokens, followed by an MLP-based diffusion head for token reconstruction. In the second phase, the masked high-resolution tokens and the predicted conditional tokens in first phase are fed into the Transformer with the Diffusion Transformer head predicting the full high-resolution token sequence. **(c) Scale-aware Transformer blocks** consist of adaLN-Zero, layernorm, self-attention, and feed-forward layers. **(d) MLP-based Diffusion head blocks** include adaLN, layernorm, and feed-forward layers. **(e) Diffusion Transformer head blocks** are composed of adaLN, layernorm, self-attention, and feed-forward layers.

all tokens to strengthen image generation. An overview of Hi-MAR is shown in Figure 2 (b).

### 3.1. Preliminary

We build our Hi-MAR upon the recent pioneering work of MAR (Li et al., 2024). Specifically, given an image $I \in \mathbb{R}^{H \times W \times 3}$, MAR first utilizes the pre-trained variational autoencoder (VAE) to encode $I$ into latent representations $I' \in \mathbb{R}^{h \times w \times d}$, where $h/w/d$ is the dimension of height/width/channels for the latent. Then, the latent $I'$ is reshaped into a sequence of $N = h \cdot w$ continuous-valued visual tokens $X = \{x_1, x_2, ..., x_N\}$. During training, MAR randomly selects $\lceil r \cdot N \rceil$ visual tokens and replaces them with masked tokens, where $r$ denotes the masking ratio sampled from a pre-defined distribution $p(r)$. The masked sequence $X' = \{x'_1, x'_2, ..., x'_N\}$ is fed into the masked autoregressive Transformer. To predict the ground-truth token $x_i$ from the masked one $x'_i$ at position $i$, a diffusion head is devised to model the probability distribution of $x_i$ conditioned on the output of the masked autoregressive Transformer $z_i$. The objective of the diffusion head is defined as the standard denoising process:

$$\mathcal{L}(z_i, x_i) = \mathbb{E}_{\varepsilon, t}\left[\left\|\varepsilon - \varepsilon_\theta(x_i^t | t, z_i)\right\|^2\right], \quad (1)$$

where $x_i^t$ is the noise-corrupted vector of $x_i$, $\varepsilon$ is the noise vector sampled from distribution $\mathcal{N}(0, \mathbf{I})$, $t$ is a time step of

the noise schedule, and $\varepsilon_\theta$ denotes the diffusion head.

Nevertheless, MAR suffers from two limitations: (1) MAR solely hinges on a single-scale sequence of visual tokens for predicting next tokens, which combines global structure construction and local details refinement into a single stage. This is contrasted to typical human perception, which first captures the global structure and then the local details in a hierarchical manner (Tian et al., 2024). (2) MAR utilizes an MLP-based diffusion head to model the masked token probability distribution individually instead of all tokens together as in conventional Diffusion Transformer (Peebles & Xie, 2023; Bao et al., 2023; Lu et al., 2024b). This way ignores the inherent structure of natural images and leaves the interdependency among all visual tokens underexploited, resulting in a sub-optimal solution for masked token prediction. Such MLP-based diffusion head could produce abnormal bright spots and fail to generate correct images (Fan et al., 2024).

### 3.2. Hierarchical Masked Autoregressive Transformer

To mitigate the first limitation of MAR, we build a top-down hierarchical masked autoregressive Transformer (Hi-MAR Transformer) with two phases. The first phase performs bidirectional autoregressive modeling over low-resolution visual tokens to capture the global structure. In the second phase, we take the output of the first phase as intermediary

pivots to guide the next autoregressive modeling over typical dense visual tokens for local details refinement. One typical variant (Tian et al., 2024) is to use a shared Transformer for the two phases and utilize the low-resolution visual tokens directly to guide the second phase prediction, as shown in Figure 2 (a). However, this training strategy often introduces a training-inference discrepancy. Specifically, during training, the models are conditioned on ground-truth low-resolution tokens $X^s$ to predict high-resolution tokens $X^l$. During inference, since ground-truth tokens are unavailable, the models have to first generate low-resolution tokens $\hat{X}^s$, which may contain errors, and then use these noisy predictions as conditions to predict the high-resolution tokens $X^l$. This mismatch between clean ground-truth low-resolution tokens $X^s$ used in training and noisy predicted low-resolution tokens $\hat{X}^s$ used in inference inevitably results in a discrepancy, resulting in performance degradation. To mitigate such training-inference discrepancy, we take the conditional tokens output from the Hi-MAR Transformer of low-resolution visual tokens for the second phase instead. The overview of our Hi-MAR is shown in Figure 2 (b). Specifically, in the first phase, the masked low-resolution visual tokens along with the context tokens (e.g., class tokens or text tokens) are fed into the Transformer, which outputs the conditional tokens $Z^s = \{z_1^s, z_2^s, ..., z_N^s\}$ on small scale. An additional diffusion head conditioned on $Z^s$ is adopted for the small scale optimization as in MAR (Li et al., 2024). In the second phase, the Transformer takes the concatenation of context tokens, small scale conditional tokens and the masked dense visual tokens as input to generate dense conditional tokens, which are further fed into Diffusion Transformer head for token prediction.

Conventional hierarchical autoregressive variants (Tian et al., 2024) model multi-scale probability distribution by a shared Transformer without additional guidance. This way could make the Transformer ambiguous and is harmful for token prediction. To alleviate the issue, we design a scale-aware Transformer block shown in Figure 2 (c). Inspired by the $adaLN\text{-}Zero$ operations adopted by DiT (Peebles & Xie, 2023), we first represent the scale information by sinusoidal embedding. The sinusoidal embedding is fed into MLP layers to generate scale vector $v$, which is leveraged to regress the scale and shift parameters of layer norm as well as the scaling parameters for residual connection. Specifically, the $i$-th scale-aware Transformer block with input $z^i$ is computed as:

$$
\begin{aligned}
\tilde{v} &= \mathbf{a} \cdot v + \mathbf{b}, \\
\alpha_1, \beta_1, \gamma_1, \alpha_2, \beta_2, \gamma_2 &= \mathbf{split}(\tilde{v}), \\
z_a &= z^i + \gamma_1 \cdot \mathbf{Attention}(\alpha_1 \cdot \mathbf{LN}(z^i) + \beta_1), \\
z^{i+1} &= z_a + \gamma_2 \cdot \mathbf{FFN}(\alpha_2 \cdot \mathbf{LN}(z_a) + \beta_2),
\end{aligned}
\tag{2}
$$

where $\mathbf{a}$ and $\mathbf{b}$ are the parameters, $\mathbf{split}$ denotes the split operation along the channel dimension, $\mathbf{LN}$, $\mathbf{Attention}$

and $\mathbf{FFN}$ denote the layernorm, self-attention and feed-forward layer, respectively. The output of the final scale-aware Transformer block acts as conditional tokens for the diffusion head.

### 3.3. Diffusion Transformer Head

To address the second limitation of MAR, we design a new Diffusion Transformer head by exploiting the self-attention to model the interdependency among tokens. In contrast to MLP-based diffusion head that only takes the conditional tokens of masked tokens as conditions, the Diffusion Transformer head considers all the masked and unmasked conditional tokens, as illustrated in Figure 2 (e). The Diffusion Transformer head contains a stack of Transformer blocks, and the $i$-th block with input $y_i$ is computed as:

$$
\begin{aligned}
\alpha_1, \beta_1, \gamma_1, \alpha_2, \beta_2, \gamma_2 &= \mathbf{split}(c), \\
y_a &= y^i + \gamma_1 \cdot \mathbf{Attention}(\alpha_1 \cdot \mathbf{LN}(y^i) + \beta_1), \\
y^{i+1} &= y_a + \gamma_2 \cdot \mathbf{FFN}(\alpha_2 \cdot \mathbf{LN}(y_a) + \beta_2),
\end{aligned}
\tag{3}
$$

where $c$ denotes the context vector obtained by summating the time step embedding and the conditional tokens, and the input of the first block is the noise-corrupted vector. Note that we only adopt Diffusion Transformer head in the second phase while the first phase still utilizes MLP-based diffusion head, since the diffusion head on the first phase mainly aims to optimize the low-resolution conditional tokens instead of providing intermediary pivots for the next phase. At inference, we use much fewer steps (e.g., 4 steps) in the second phase considering that the Diffusion Transformer head is much heavier than the MLP-based diffusion head. With the global structure provided by the first phase, the second phase can focus on the local fine-grained details and requires much fewer steps to generate satisfied results.

## 4. Experiments

### 4.1. Datasets

We empirically verify the merit of hierarchical masked autoregressive models for image generation in comparison with state-of-the-art approaches on two datasets, i.e., ImageNet (Deng et al., 2009) and MS-COCO (Lin et al., 2014). For class-conditional image generation, we validate Hi-MAR on ImageNet at $256 \times 256$ resolution, which consists of 1,281,167 training images from 1K different classes. For text-to-image generation, we evaluate Hi-MAR on MS-COCO at $256 \times 256$, which is composed of 82,783 training images and 40,504 validation images. Each image is annotated with five captions. Following Stable Diffusion (Rombach et al., 2022), we convert captions into a sequence of text embeddings with CLIP text encoder. Then the text embeddings act as context tokens and are fed into Hi-MAR for autoregressive modeling.

*Table 1.* The architecture configurations of the family of Hi-MAR in three different scales (i.e., Base, Large, and Huge). Diff. Head$_{1/2}$ denotes the diffusion head on the first/second phase.

| Model | Hi-MAR Transformer | | Diff. Head$_1$ | | Diff. Head$_2$ | | #Params |
|-------|---------|-------------|---------|-------------|---------|-------------|---------|
| | #Layers | Hidden size | #Layers | Hidden size | #Layers | Hidden size | |
| Hi-MAR-B | 24 | 768 | 6 | 1024 | 6 | 512 | 244M |
| Hi-MAR-L | 32 | 1024 | 8 | 1280 | 8 | 512 | 529M |
| Hi-MAR-H | 40 | 1280 | 12 | 1536 | 12 | 768 | 1090M |

## 4.2. Experimental Settings

**Image Tokenizer.** We employ the variational autoencoder (KL-16 version) trained by MAR (Li et al., 2024) to encode low-resolution (128×128) and high-resolution (256×256) images into latent representations for the two phases.

**Network Architectures.** Following the architecture configurations of MAR family (Li et al., 2024), we build three variants of our Hi-MAR in three different scales (i.e., Base, Large and Huge). To compare with MAR-B/L/H, the number of Transformer blocks in masked autoregressive Transformer of Hi-MAR-Base (Hi-MAR-B), Hi-MAR-Large (Hi-MAR-L), and Hi-MAR-Huge (Hi-MAR-H) are set as 24,32, and 40 respectively. The number of Transformer blocks in the diffusion head on both phases of Hi-MAR-B/L/H is 6/8/12. Table 1 shows the detailed configurations (e.g., layer number, hidden size) of three Hi-MAR variants.

**Training Setup.** At training stage, we conduct all experiments on 80GB-H100 GPUs. For class-conditional image generation on ImageNet, we follow MAR (Li et al., 2024) and train the models using AdamW optimizer ($\beta_1 = 0.9, \beta_2 = 0.95$) with 0.02 weight decay for 800 epochs. We use the constant lr schedule with a 1e-4 learning rate and 100-epoch linear warmup. In the first phase, the masking ratio is randomly sampled in $[0.7, 1.0]$ as MAR, while the second phase uses the cosine masking strategy following MaskGIT (Chang et al., 2022). For text-to-image generation on MS-COCO, we follow AutoNAT-L (Ni et al., 2024) and randomly sample the masking ratio by Beta distribution ($\alpha = 4, \beta = 1$). The AdamW optimizer is adopted with an 8e-4 learning rate, 0.03 weight decay and 8K-step linear warmup. The exponential moving average is adopted with a momentum of 0.9999. At inference, we use 32 and 4 steps for the first and second phases with a cosine schedule.

**Evaluation Metrics.** For evaluation, we use Fréchet Inception Distance (FID) (Heusel et al., 2017), Inception Score (IS) (Salimans et al., 2016) and Precision/Recall (Kynkäänniemi et al., 2019) on 50K generated samples to measure the image quality on ImageNet. On MS-COCO, we randomly draw 30K prompts from the validation set and generate samples on these prompts as U-ViT (Bao et al., 2023). We report the FID score as the main metric.

## 4.3. Results on Class-Conditional Image Generation

Table 2 summarizes the performance comparisons of different methods for class-conditional image generation on ImaegNet dataset. All runs are grouped into four categories: the Generative Adversarial Network(GAN) based models, diffusion-based models, autoregressive models, and masked autoregressive models. Except for the GAN based models, we report the performances of the runs under two different inference settings i.e., with or without Classifier-Free Guidance (CFG) (Ho & Salimans, 2021). For our method under the w/o CFG setting, the CFG is only turned off during the prediction of dense tokens, as the first-stage output quality significantly affects Hi-MAR performance.

Overall, under the two different settings, the results across most metrics consistently indicate that our Hi-MAR achieves superior performances against the state-of-the-art models among all the four categories with comparable parameter size. In particular, the FID score of Hi-MAR-B on the base scale achieves 1.93 under CFG, making the absolute improvement over the best competitor MAR-B by 0.38. The results generally highlight the key advantage of exploiting hierarchical autoregressive and modeling interdependency among tokens. Specifically, U-ViT and DiT upgrade the conventional U-Net structure diffusion model with Transformer, resulting in remarkable scaling property with superior performances than all GAN based and U-Net structure diffusion models. Note that the use of CFG generally improves the FID, IS and Precision scores for diffusion and AR models across different scales. But the Recall scores decrease due to CFG tends to improve the generation quality at the cost of sacrificing diversity. Compared to diffusion models, autoregressive models (e.g., VQGAN (Esser et al., 2021) and LlamaGen (Sun et al., 2024)) regard images as a sequence of discrete tokens, achieving comparable image generation results. Furthermore, the mask autoregressive models introduce mask tokens into autoregressive modeling, facilitating bidirectional contextual information learning along generative modeling, and thus improve performance. Instead of quantizing images into discrete tokens, MAR utilizes a continuous tokenizer via the diffusion loss, leading to significant performance boosts. But the performance of MAR across different model sizes is still inferior to our Hi-MAR. This validates the effectiveness of hierarchical autoregressive modeling and the Diffusion Transformer head for enhanced masked token prediction with richer context information among all tokens.

## 4.4. Results on Text-to-Image Generation

Table 3 shows the performance comparison on MS-COCO for text-to-image generation. For fair comparison, we follow the configuration of U-ViT-S/2 (Deep) (Bao et al., 2023) and build a light-weight version of our Hi-MAR with compara-

*Table 2.* **Generative model family comparison on class-conditional ImageNet 256×256**. "↓" or "↑" indicate lower or higher values are better. Metrics include Fréchet inception distance (FID), inception score (IS), precision and recall. Models with the suffix "-re" used rejection sampling.

| Type | Model | #Para. | FID↓ | IS↑ | Precision↑ | Recall↑ | FID↓ | IS↑ | Precision↑ | Recall↑ |
|------|-------|--------|------|-----|------------|---------|------|-----|------------|---------|
| | | | | w/o CFG | | | | w/ CFG | | |
| GAN | BigGAN (Brock et al., 2019) | 112M | 6.95 | 224.5 | **0.89** | 0.38 | – | – | – | – |
| | GigaGAN (Kang et al., 2023) | 569M | 3.45 | 225.5 | 0.84 | 0.61 | – | – | – | – |
| | StyleGan-XL (Sauer et al., 2022) | 166M | 2.30 | 265.1 | 0.78 | 0.53 | – | – | – | – |
| Diff. | ADM (Dhariwal & Nichol, 2021) | 554M | 10.94 | 101.0 | 0.69 | 0.63 | 4.59 | 186.7 | 0.82 | 0.52 |
| | CDM (Ho et al., 2022) | – | – | – | – | – | 4.88 | 158.7 | – | – |
| | LDM-4-G (Rombach et al., 2022) | 400M | 10.56 | 103.5 | 0.71 | 0.62 | 3.60 | 247.7 | **0.87** | 0.48 |
| | U-ViT-H/2 (Bao et al., 2023) | 501M | – | – | – | – | 2.29 | 263.88 | 0.82 | 0.57 |
| | DiT-XL/2 (Peebles & Xie, 2023) | 675M | 9.62 | 121.5 | 0.67 | 0.67 | 2.27 | 278.2 | 0.83 | 0.57 |
| AR | VQGAN (Esser et al., 2021) | 227M | 18.65 | 80.4 | 0.78 | 0.26 | – | – | – | – |
| | VQGAN-re (Esser et al., 2021) | 1.4B | 5.20 | 280.3 | – | – | – | – | – | – |
| | RQTran. (Lee et al., 2022) | 3.8B | – | – | – | – | 7.55 | 134.0 | – | – |
| | RQTran.-re (Lee et al., 2022) | 3.8B | – | – | – | – | 3.80 | **323.7** | – | – |
| | GIVT (Tschannen et al., 2025) | 304M | 5.67 | – | 0.75 | 0.59 | 3.35 | – | 0.84 | 0.53 |
| | LlamaGen-L (Sun et al., 2024) | 343M | 19.07 | 64.3 | 0.61 | 0.67 | 3.07 | 256.06 | 0.83 | 0.52 |
| | LlamaGen-XL (Sun et al., 2024) | 775M | 15.54 | 79.2 | 0.62 | **0.69** | 2.62 | 244.08 | 0.80 | 0.57 |
| | LlamaGen-XXL (Sun et al., 2024) | 1.4B | 14.65 | 86.3 | 0.63 | 0.68 | 2.34 | 253.90 | 0.80 | 0.59 |
| | VAR-d16 (Tian et al., 2024) | 310M | – | – | – | – | 3.30 | 274.4 | 0.84 | 0.51 |
| | VAR-d20 (Tian et al., 2024) | 600M | – | – | – | – | 2.57 | 302.6 | 0.83 | 0.56 |
| | VAR-d24 (Tian et al., 2024) | 1.0B | – | – | – | – | 2.09 | 312.9 | 0.82 | 0.59 |
| Mask. | MaskGIT (Chang et al., 2022) | 227M | 6.18 | 182.1 | 0.80 | 0.51 | – | – | – | – |
| | AutoNAT-L (Ni et al., 2024) | 422M | – | – | – | – | 2.68 | 278.8 | – | – |
| | MAR-B (Li et al., 2024) | 208M | 3.48 | 192.4 | 0.78 | 0.58 | 2.31 | 281.7 | 0.82 | 0.57 |
| | MAR-L (Li et al., 2024) | 479M | 2.60 | 221.4 | 0.79 | 0.60 | 1.78 | 296.0 | 0.81 | 0.60 |
| | MAR-H (Li et al., 2024) | 943M | 2.35 | 227.8 | 0.79 | 0.62 | 1.55 | 303.7 | 0.81 | 0.62 |
| Hi-MAR | Hi-MAR-B | 244M | 2.11 | 251.46 | 0.80 | 0.59 | 1.93 | 293.0 | 0.81 | 0.59 |
| | Hi-MAR-L | 529M | 1.72 | 278.63 | 0.79 | 0.62 | 1.66 | 322.3 | 0.79 | 0.61 |
| | Hi-MAR-H | 1090M | **1.55** | **300.72** | 0.80 | 0.63 | **1.52** | 322.78 | 0.80 | **0.63** |

*Table 3.* **FID results of different models on MS-COCO 256 × 256 validation.** "↓" indicates lower values are better.

| Type | Model | FID ↓ |
|------|-------|-------|
| GAN | AttnGAN (Xu et al., 2018) | 35.49 |
| | DM-GAN (Zhu et al., 2019) | 32.64 |
| | DF-GAN (Tao et al., 2022) | 19.32 |
| | XMC-GAN (Zhang et al., 2021) | 9.33 |
| | LAFITE (Zhou et al., 2022) | 8.12 |
| Diffusion | VQ-Diffusion (Gu et al., 2022) | 19.75 |
| | Friro (Fan et al., 2023) | 8.97 |
| | U-ViT-S/2 (Deep) | 5.48 |
| Mask. | AutoNAT-S (Ni et al., 2024) | 5.36 |
| | MAR (Li et al., 2024) | 6.36 |
| Hi-MAR | Hi-MAR-S | **4.77** |

ble model size. We group all runs in three categories, i.e., GAN based models, diffusion models, and masked autoregressive models. In general, our Hi-MAR outperforms other baselines on this challenging dataset. In particular, the FID score of Hi-MAR can achieve 4.77, making the absolute improvement over the best competitor AutoNAT-S by 0.59.

Similar to the observation in ImageNet, diffusion models with convolutional structure generally exhibit more flexible and scalable generative modeling and thus achieve better performance. U-ViT further obtains better results by replacing convolutional structure with Transformer, basically validating the effectiveness of Diffusion Transformer as a higher-capacity backbone. There is a large performance gap between MAR and our Hi-MAR. Though both runs belong to masked autoregressive models with continuous tokenizer, Hi-MAR upgrades MAR with hierarchical masked autoregressive modeling that creates images from global structure to local details and Diffusion Transformer head to mine context among all tokens, yielding apparent improvements.

Furthermore, we assess the compositional alignment between generated images and input text using T2I-CompBench (Huang et al., 2023), a comprehensive benchmark designed to evaluate fine-grained compositional understanding in text-to-image generation. We compare our Hi-MAR with other state-of-the-art methods which are trained on MS-COCO with similar parameter size. As shown in Table 4, Hi-MAR outperforms existing baselines across several crucial aspects, including attribute binding, object

*Table 4.* **T2I-CompBench evaluation of different models.** "↑" indicates higher values are better.

| Model | Attribute Binding | | | Object Relationship | | Complex ↑ |
|---|---|---|---|---|---|---|
| | Color ↑ | Shape ↑ | Texture ↑ | Spatial ↑ | Non-Spatial ↑ | |
| U-ViT-S/2 (Deep) (Bao et al., 2023) | 0.3626 | 0.2682 | 0.3474 | 0.0353 | **0.2693** | 0.2219 |
| AutoNAT-S (Ni et al., 2024) | 0.3225 | 0.2466 | 0.3389 | 0.0453 | 0.2468 | 0.2024 |
| Hi-MAR-S | **0.3862** | **0.2782** | **0.3945** | **0.0409** | 0.2690 | **0.2313** |

*Table 5.* **Ablation study of Hi-MAR.** Pivots denote the first phase generated tokens that act as additional guidance for the next phase. Diff. Head$_{1/2}$ denotes the diffusion head on the first/second phase.

| Pivots | Diff. Head$_1$ | Diff. Head$_2$ | Scale vector | #Para. | FID↓ |
|---|---|---|---|---|---|
| ✗ | ✗ | MLP-based | ✗ | 208M | 2.31 |
| visual tokens | MLP-based | MLP-based | ✗ | 245M | 2.28 |
| conditional tokens | MLP-based | MLP-based | ✗ | 245M | 2.07 |
| conditional tokens | MLP-based | Transformer | ✗ | 239M | 1.98 |
| conditional tokens | Transformer | Transformer | ✗ | 233M | 1.98 |
| conditional tokens | MLP-based | Transformer | ✓ | 242M | **1.93** |

relationships, and complex compositions, demonstrating its capability to generate semantically aligned images.

### 4.5. Experimental Analysis

**Ablation Study.** Here we study how each design in our Hi-MAR framework influences the overall performance. Recall that our Hi-MAR upgrades typical MAR with three novel designs, i.e., Hi-MAR Transformer that creates images from global structure to local details, scale-aware Transformer block that provides scale guidance to the Transformer on the two phases, and Diffusion Transformer head that models the interdependency among tokens and strengthens masked token prediction. Table 5 details the performance across different ablated runs of Hi-MAR on ImageNet for class-conditional image generation. In particular, we start from the base model of typical MAR (the first row in this table), which enables masked autoregressive modeling through a diffusion loss within a continuous-valued space. By incorporating Hi-MAR Transformer and using low-resolution visual tokens to guide the second phase prediction (the second row), FID only improves 0.03 due to the discrepancy between training and inference as mentioned in Section 3.2. The masked autoregressive Transformer relies too much on the low-resolution tokens and tends to merely resize them into larger resolution. Thus, it fails to correct the flawed tokens predicted by the first phase. When adopting the conditional tokens to mitigate such discrepancy (the third row), the FID notably improves over MAR by 0.24. Next, we upgrade the MLP-based diffusion head to Diffusion Transformer head (the fourth row) for the second phase to model interdependency among tokens, the FID score further improves to 1.98. Nevertheless, there is no large improvement when the diffusion head in the first phase is replaced by Diffusion Transformer head (the fifth row). The full version

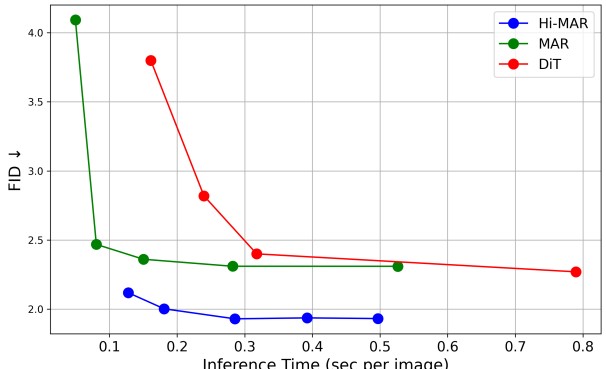

*Figure 3.* **Speed/accuracy trade-off.**

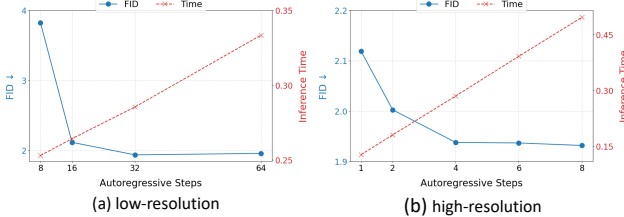

*Figure 4.* **Impact of autoregressive steps.** The experiments are conducted using the Hi-MAR-B model. For experiments varying low-resolution inference steps, typical dense inference steps are fixed at 4. Similarly, for experiments varying typical dense inference steps, low-resolution inference steps are fixed at 32.

of our Hi-MAR (the last row) is finally benefited from our three key designs, and achieves the best performances across most metrics. The results basically validate our design.

**Qualitative Results.** To qualitatively evaluate our Hi-MAR, we showcase 24 image results generated by MAR and Hi-MAR on ImageNet and MS-COCO in figure Figure 5. We clearly observe that Hi-MAR generates higher-quality images with less distortions and better aligned semantics with input class/caption, validating the effectiveness of exploiting hierarchical autoregressive and modeling interdependency among tokens.

**Speed/accuracy Trade-off.** Following MAR (Li et al., 2024), we plot the speed/accuracy trade-off curves of DiT-XL/2, MAR-B and Hi-MAR-B in figure Figure 3. The curve of DiT-XL/2 is obtained by different diffusion steps (50, 75, 100, 250), while the curve of MAR-B is measured by different autoregressive steps (16, 32, 64, 128, 256). For Hi-MAR-B, we fix the steps on the first phase as 32 and varying the number of steps on the second phase (1,2,4,6,8). We measure the speed on ImageNet 256×256 using one H100 GPU with batch size 128. It is clear that our Hi-MAR has a better trade-off than MAR and DiT-XL/2.

**Impact of Autoregressive Steps.** We analyze the impact of autoregressive steps on both phases. As shown in figure

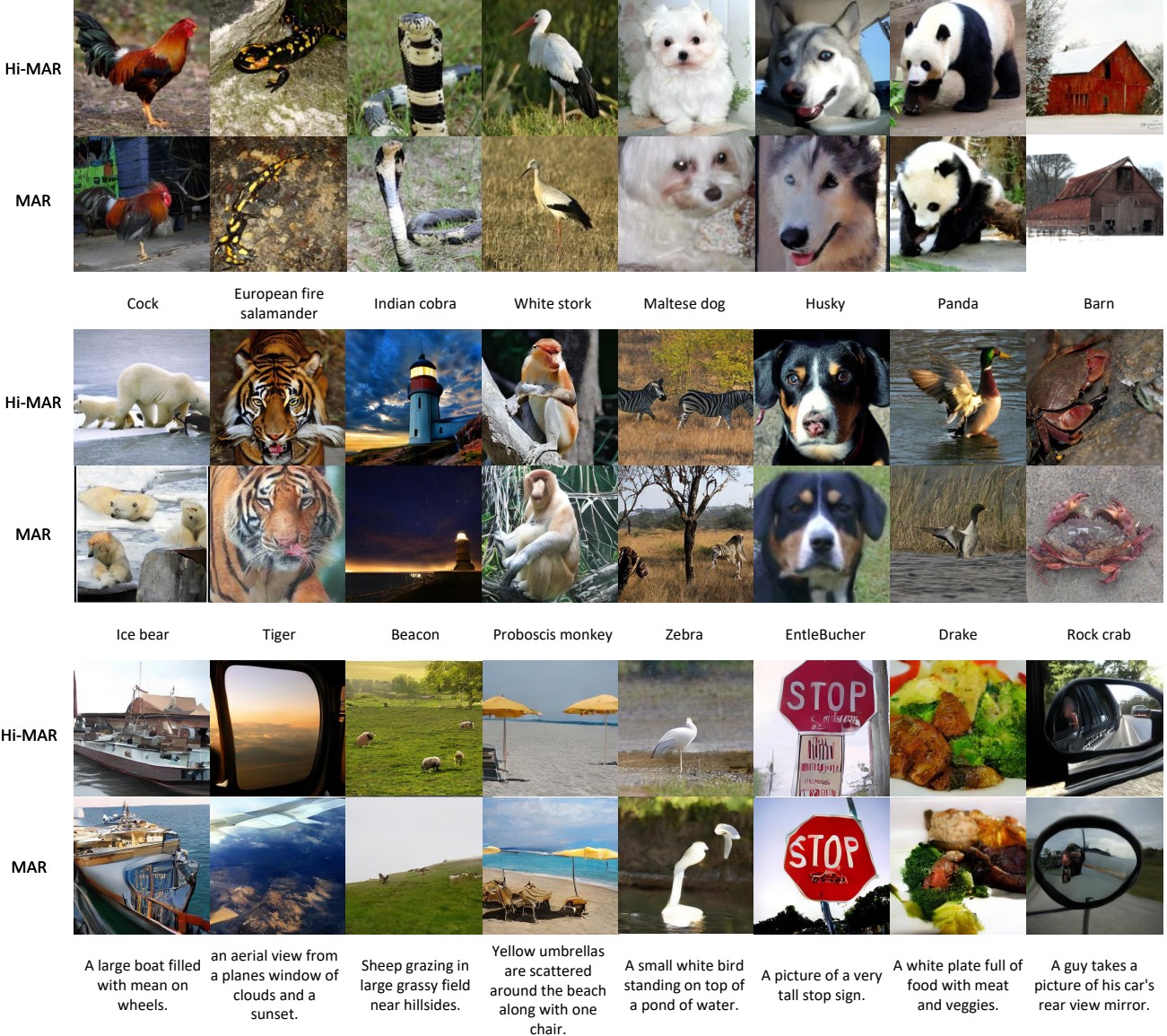

*Figure 5.* **Qualitative results on class-conditional image generation and text-to-image generation.** The top two rows show class-conditional generation, while the bottom row show text-to-image generation.

Figure 4 (a), the FID decreases as the step number on the first phase increases and reaches an optimal value at 32 steps. With the global structure provided by the first phase, the second phase can focus more on the fine-grained local details and manage to achieve nearly saturated FID with just 4 autoregressive steps, as illustrated in figure Figure 4 (b). Therefore, we set the number of autoregressive steps on both phases as 32 and 4 for better trade-off between generation quality and inference speed.

## 5. Conclusion

In this work, we discuss the limitations of next-token prediction in visual autoregressive modeling caused by the lack of global context. To overcome this, we propose a new hierarchical autoregressive prediction framework that establishes a hierarchy from low-resolution image tokens, which capture global structure with coarse details, to high-resolution image tokens, which provide fine-grained details. Building on this framework, we introduce the Hierarchical Masked Autoregressive Model (Hi-MAR), which demonstrates superior performance compared to widely-used diffusion models and conventional autoregressive baselines. We hope that our work will highlight the importance of utilizing global context in visual autoregressive modeling and hope it inspires further research in this direction.

## Acknowledgments.

This work was supported in part by the Beijing Municipal Science and Technology Project No. Z241100001324002, Beijing Nova Program No. 20240484681 and National Natural Science Foundation of China (No. 62332016) and the Key Research Program of Frontier Sciences, CAS (No. ZDBS-LY-JSC001).

## Impact Statement

This paper presents work whose goal is to advance the field of Machine Learning. There are many potential societal consequences of our work, none which we feel must be specifically highlighted here.

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
