# OpenReview forum: "Hierarchical Masked Autoregressive Models with Low-Resolution Token Pivots"
_ICML.cc/2025/Conference — ICML 2025 poster_

### Official Review · Reviewer_dCxF · 2025-03-03

**Overall Recommendation:** 3

**Summary:**

This paper proposed a Hierarchical Masked Autoregressive Model based on MAR (Li et al., 2024) by introducing a low-resolution modeling phase, which is claimed to provide global structure guidance for generating dense image tokens. Specifically, x2-lower-res image tokens are first modeled by a scale-aware transformer block in a bi-directional masked-modeling manner, followed by a MLP diffusion head to predict the continuous latents. Then, at the second phase, the predicted latents, rather than the ground-truth tokens, are used as additional conditions for generating high-res tokens with the same scale-aware transformer but a different transformer-based diffusion head. The proposed model is tested on ImageNet and MS-COCO for C2I and T2I generation and compared with existing approaches.

**update after rebuttal**: Based on the reviews, rebuttal, and discussions, I find this paper to be borderline. While the additional information and experiments provided in the rebuttal help strengthen the paper within its reasonable scope, I remain unconvinced by its level of novelty. That said, I have raised my final rating to a weak accept.

**Claims And Evidence:**

The essential claims, including the lack of global context, training-inference discrepancy, independent sampling issue, and speed/accuracy trade-off, are well-justified and largely supported by empirical results within the scope of this paper.

However, whether these claims are conclusive for large-scale problems or models is unclear. For example,
  - The **training-inference discrepancy** issue is only addressed for passing tokens from phase 1 to phase 2, while during the autoregressive process of stage 1 and stage 2, teacher-forcing learning is still applied.
  - It is unclear how **speed and accuracy** behave for longer visual sequences. For example, will the additional phase 1 and the Diffusion Transformer head in phase 2 lead to much higher computational cost for higher-res images?

**Essential References Not Discussed:**

This paper includes and clearly refers to most of the essential literature, and I have no problem with this. There are many concurrent works about AR+continuous latent and AR+multi-scale (e.g., Fluid, Infinity, FlowAR, HART, FlexVAR, FractalAR, ...) that can be added to the later version of this paper.

**Experimental Designs Or Analyses:**

1. I am not satisfied with the T2I experiment. The model is only tested on MS-COCO and only compares FID, which can hardly reflect the actual performance.
 - Other more comprehensive benchmarks should be evaluated, such as T2I-CompBench (Huang et al., 2023) and GenEval (Ghosh et al., 2023).
 - Only a small-scale model (Hi-MAR-S) is tested. By the way, the exact configuration of Hi-MAR-S is not specified in Table 1.

2. Is the diffusion loss at the second stage the only training objective? How many steps are the Diffusion heads trained for? Is it the same as MAR (1000 steps)? And do the models in Figure 4 follow this training setup?

3. Table 2 (w/ CFG) shows that as the model size increases, the gap between Hi-MAR and MAR reduces. I am concerned that the proposed method is not scalable to larger images and large models.

**Methods And Evaluation Criteria:**

The proposed method is largely based on MAR (and incorporates many ideas from the VAR paper about scales). It is reasonable, well-motivated, and clearly feasible.

The paper follows MAR and uses the most widely applied ImageNet dataset to evaluate 256x256-res C2I generation. The paper also uses MS-COCO to evaluate T2I results. Both are very common practices and reasonable.

**Other Comments Or Suggestions:**

At the current stage, I think the most doable items are expanding the T2I experiment, providing more visualizations, and improving the analysis to include more insightful studies, such as the impact on phase 1 resolution and other design choices of the diffusion heads.

To strengthen this paper, I expect experiments on longer visual sequences (higher-resolution images) and larger models to justify the proposed method's scalability, efficiency, and generalization ability.

**Other Strengths And Weaknesses:**

I like the overall idea of introducing scale-wise modeling to MAR, along with other nice adaptations. To me, this is a safe innovation but somewhat incremental. My major concern is that the scope and depth of this study is too small or shallow to reveal the potential of the approach.

**Questions For Authors:**

I have specified my concerns and questions in the sections above, please refer to those parts.

**Relation To Broader Scientific Literature:**

The proposed method is largely based on MAR (Li et al., 2024) and incorporates many ideas about scales from the VAR (Tian et al., 2024) paper. In short, MAR proposes a diffusion head based on masked-modeling bidirectional autoregressive model for predicting continuous latents instead of discrete code. VAR proposes a scale-wise instead of token-wise autoregressive paradigm. The key contribution of this paper, introducing an extra smaller-scale autoregressive modeling phase to the single-stage MAR framework, is highly relevant to the VAR's scale-wise idea.

**Theoretical Claims:**

The essential claims, including the lack of global context, training-inference discrepancy, independent sampling issue, and speed/accuracy trade-off, are well-justified and *largely supported* by empirical results *within the scope of this paper*. I have mentioned potential concerns in Claims And Evidence and Experimental Designs Or Analyses in my review, please refer to those sections for details.

---

> ### Author Rebuttal · Authors · 2025-04-01
>
> **Q1: Training-inference discrepancy issue**
>
> Yes. We mainly focus on the design of hierarchical masked autoregressive model which addresses the training-inference discrepancy for passing tokens from phase 1 to phase 2, while the discrepancy caused by the inherent autoregressive process in each stage still remains. Such discrepancy is also occurred on most existing autoregressive models. We will discuss this.
>
> **Q2: Speed and accuracy behave for longer visual sequences**
>
> As suggested, we experimented on larger 512 resolution and Hi-MAR-L (FID: 1.62) outperforms MAR-L (FID: 1.73), while its computational cost is 20.9% less than MAR-L. We will add this.
>
> **Q3: More comprehensive benchmarks**
>
> Thanks. As suggested, we evaluate T2I model on T2I-CompBench and GenEval benchmarks. Due to limited computational resources, here we compare our Hi-MAR with other state-of-the-art methods (e.g., U-ViT-S and AutoNAT-S) which are trained on MS-COCO with similar parameter size. The larger models with billions of parameters trained on billions of images (e.g., SDXL, SD3) are not included. As shown in the following tables, our Hi-MAR consistently outperforms other baselines with comparable parameter size. We will add this.
>
> | Method    | Single Obj. | Two Obj. | Counting | Colors | Positions | Color Attri. | Overall |
> | --------- | ----------- | -------- | -------- | ------ | --------- | ------------ | ------- |
> | U-ViT-S   |   83.75     |  18.69   |  16.25   | 38.03  |  3.00     |   0.75       |  26.75  |
> | AutoNAT-S |   81.25     |  18.18   |  16.56   | 36.70  |  3.75     |   1.25       |  26.28  |
> | Hi-MAR-S  |   89.06     |  22.73   |  17.50   | 44.41  |  2.50     |   2.25       |  29.74  |
>
> | Method    | Color  | Shape  | Texture | Spatial | Non-Spatial | Complex |
> | --------- | ------ | ------ | ------- | ------- | ----------- | ------- |
> | U-ViT-S   | 0.3626 | 0.2682 | 0.3474  | 0.0353  | 0.2693      | 0.2219  |
> | AutoNAT-S | 0.3225 | 0.2466 | 0.3389  | 0.0453  | 0.2468      | 0.2024  |
> | Hi-MAR-S  | 0.3862 | 0.2782 | 0.3945  | 0.0409  | 0.2690      | 0.2313  |
>
> **Q4: Configuration of Hi-MAR-S**
>
> For fair comparison, Hi-MAR-S follows the configuration of U-ViT-S/2 (Deep) and its exact configuration is shown in the following table. We will add this in Table 1.
>
> |          |         | Hi-MAR Transformer |         | Diff. Head1 |         | Diff. Head2 |         |
> | -------- | ------- | ------------------ | ------- | ----------- | ------- | ----------- | ------- |
> | Model    | #Layers | Hidden size        | #Layers | Hidden size | #Layers | Hidden size | #params |
> | Hi-MAR-S | 17      | 512                | 5       | 512         | 5       | 512         | 108M    |
>
> **Q5: Training setup**
>
> To be clear, we only employ the diffusion loss as the training objective on both stages. Following MAR, we set the maximum timestep as 1,000 for the Diffusion heads. The models in Figure 4 follow this training setup.
>
> **Q6: The gap between Hi-MAR and MAR reduces. The scalablility of Hi-MAR to larger images and larger models**
>
> The performance in ImageNet is almost saturated and it is relatively difficult to introduce large margin of improvement. Considering this comment, we conducted the suggested experiments on larger resolution (i.e., 256 -> 512) and the FID achieves 1.62, which improves over MAR-L by 0.11. Furthermore, we scale both MAR and Hi-MAR to 2B parameters, and the FID of MAR and Hi-MAR is 1.49 and 1.45 respectively. The results basically demonstrate that Hi-MAR is scalable to both larger images and larger models. We will add this.
>
> **Q7: Concurrent works**
>
> We appreciate the suggested concurrent works and we are also happy to discuss them in revised version.
>
> **Q8: Safe innovation**
>
> Please refer to Q1 of Reviewer SRgF for more discussion on technical contribution against existing works.
>
> **Q9: More visualizations**
>
> As suggested, we provide more visualization results on the [link](https://anonymous.4open.science/r/HiMAR_Visual/README.md). We will add this.
>
> **Q10: Impact on phase 1 resolution**
>
> We experimented with smaller resolution (i.e., 64x64) on phase 1 and the FID score is degraded into 2.06 due to the quality of images generated in such smaller resolution is relatively low. Therefore, we choose 128x128 resolution for the first phase. We will add this.
>
> | Phase 1 Resolution | FID  |
> | ------------------ | ---- |
> | 64x64              | 2.06 |
> | 128x128            | 1.93 |
>
> **Q11: Other design choices of the diffusion heads**
>
> We also experimented by replacing the self-attention layer with cross-attention in the Diffusion Transformer head to mine the context among all tokens, and the FID is slightly dropped by 0.05 compared to the final version of Hi-MAR. We will discuss this in ablation study.

---

> > ### Comment · Reviewer_dCxF · 2025-04-05
> >
> > Thanks to the authors for responding to my questions and providing additional results that support the claims more. I don't have any further questions, and I will decide the final rating based on all the reviews, rebuttals, and discussions. Thanks.

---

### Official Review · Reviewer_SRgF · 2025-03-15

**Overall Recommendation:** 3

**Summary:**

This paper proposes a hierarchical masked AR visual generation model with low-resolution tokens as pivots. By first generating low-resolution image tokens, which provide a global structure, the second generation phase can benefit from the global context. Besides, the proposed diffusion transformer head to further improve the results. Experimental results show that Hi-MAR can obtain better performance than baselines.

**Claims And Evidence:**

Yes

**Essential References Not Discussed:**

See Relation To Broader Scientific Literature.

**Experimental Designs Or Analyses:**

Yes.

**Methods And Evaluation Criteria:**

Yes

**Other Comments Or Suggestions:**

N/A

**Other Strengths And Weaknesses:**

Strengths:
1. The paper is easy to follow, and the idea is intuitively easy to understand.
2. The performance is impressive.
3. Reduce the ar steps in the second step and improve the generation speed.
Weakness:
1. Limited novelty. See Relation To Broader Scientific Literature
2. Missing speed comparison with VAR and HART.

**Questions For Authors:**

1. Have you tried other resolution settings, like 128->512,  256->512?

**Relation To Broader Scientific Literature:**

I think the key problem with Hi-Mar is the novelty improvement with Muse[1], VAR[2], and Hart[3].
1. Muse also uses a super-resolution transformer to generate the final image with low-resolution information fused by cross-attention.
2. VAR proposes to use multi-scale generation to image generation. Hi-Mar only uses one low-resolution scale. I think this is a special case  for VAR.
3. In Hart (missing reference), they use residual diffusion to improve the performance of VAR.
Considering these three papers, the novelty of Hi-Mar is limited enough.


[1] Muse: Text-to-image generation via masked generative transformers. ICML 2023.
[2] Visual Autoregressive Modeling: Scalable Image Generation via Next-Scale Prediction. NeurIPS 2024.
[3] HART: Efficient Visual Generation with Hybrid Autoregressive Transformer. ICLR 2025.

**Theoretical Claims:**

No theoretical claims

---

> ### Author Rebuttal · Authors · 2025-04-01
>
> **Q1: Novelty**
>
> Thanks. We summarize the differences between Hi-MAR and conventional multi-scale generation models as follows:
>
> 1) During training, the conventional models (i.e., Muse, VAR, Hart) commonly utilizes the ground-truth low-resolution visual tokens directly to guide the next phase prediction. Instead, Hi-MAR takes the conditional tokens estimated from the Hi-MAR Transformer of low-resolution visual tokens as the condition. Such design can mitigate the training-inference discrepancy as discussed in section 3.2. As shown in Table 4, the FID score improves from 2.28 to 2.07 when adopting the conditional tokens to mitigate such discrepancy.
> 2) Both VAR and Hart models multi-scale probability distribution via a shared Transformer without additional guidance. This way leaves the inherent different peculiarities of each scale in autogressive modeling not fully exploited, resulting in a sub-optimal solution for multi-scale token prediction. In contrast, our Hi-MAR incorporates a scale-aware Transformer block that provides scale guidance to the Transformer tailored to each phase.
> 3) Both VAR and Muse discretize images by VQGAN, resulting in severe information loss. Instead, Hi-MAR adopts continuous tokenizer via a diffusion loss, overcoming the poor generation upper bound caused by the vector quantization.
> 4) Muse utilizes two different models for low/high-resolution image generation and the two models are trained separately. Instead, our Hi-MAR jointly optimizes the probability distribution for low/high-resolution tokens with a shared scale-aware masked autoregressive Transformer and two small diffusion heads, which is more parameter-efficient.
> 5) In contrast to Hart that utilizes an MLP-based diffusion head to model the each token probability distribution individually, Hi-MAR devises Diffusion Transformer head by exploiting the self-attention to model the interdependency among tokens. Note that we appreciate the suggested reference of concurrent work Hart (pulished on ICLR 2025 with camera ready deadline on Mar 14, 2025). We will add the discussion.
>
>
>
>
> **Q2: Speed comparison with VAR and HART**
>
> As suggested, we compare the speed of our Hi-MAR with mentioned VAR and HART in the following table. The results basically demonstrate that Hi-MAR achieves superior performance against VAR and HART with comparable computational costs. We will add this in revision.
>
> | Method   | #Para. | FID  | Phase_1 Steps     | Phase_2 Steps  | Diff. Head_1 Step | Diff. Head_2 Step | Inference Time/per image |
> | -------- | ------ | ---- | ------------ | ------------ | ------------------------ | ------------------------ | ------------------------ |
> | VAR-d20  | 600M   | 2.57 | 10           | &cross;      | &cross;                  | &cross;                  | 0.14489                  |
> | HART-d20 | 649M   | 2.39 | 10           | &cross;      | 8                        | &cross;                  | 0.15401                  |
> | DiT-XL/2 | 675M   | 2.27 | 250          | &cross;      | &cross;                  | &cross;                  | 0.78970                  |
> | MAR-B    | 208M   | 2.31 | 256          | &cross;      | 100                      | &cross;                  | 0.52641                  |
> | Hi-MAR-B | 244M   | 2.00 | 32           | 4            | 100                      | 50                       | 0.13587                  |
> | Hi-MAR-B | 244M   | 1.93 | 32           | 4            | 100                      | 250                      | 0.28552                  |
>
> **Q3: Other resolution settings**
>
> Thanks. We experimented by equipping Hi-MAR-L with larger resolution (i.e., 256 -> 512), and the FID score achieves 1.62, which outperforms MAR-L with 512 resolution by 0.11. The result again validates the effectiveness of exploiting hierarchical autoregressive and modeling interdependency among tokens. We will add this.

---

### Official Review · Reviewer_kLHw · 2025-03-15

**Overall Recommendation:** 3

**Summary:**

The paper introduces Hi-MAR, a hierarchical masked generative model for visual generation. Hi-MAR first predicts low-resolution image tokens as global structural pivots, which then guide the next phase of dense token prediction, enhanced by a Diffusion Transformer head for better global context modeling. Experiments on image generation tasks show that it outperforms baselines and is computationally efficient.

**Claims And Evidence:**

Fine. The claims are intuitive and easy to follow.

**Essential References Not Discussed:**

N/A

**Experimental Designs Or Analyses:**

Given the success of recent next-scale prediction approaches, such as VAR, it is not surprising that the proposed cascaded method could be beneficial.
However, its practical advantage over VAR remains unclear.
Additionally, the method explicitly adopts a "two-stage" approach—would further stages yield additional improvements?
More experiments and analyses may be required to address the two concerns.

**Methods And Evaluation Criteria:**

Fine.

**Other Comments Or Suggestions:**

Has the autoregressive relationship in Figure 1(a) been drawn incorrectly? Is the causal relationship represented by the black connecting lines reversed? It seems like a left-right mirror flip of the figure would be correct.

##### Post-rebuttal: Thanks to the authors for the detailed responses. I am ok with the rebuttal. I hope the discussions can be incorporated into the revision. Good luck.

**Other Strengths And Weaknesses:**

The reviewer acknowledges that the paper's technical contributions to visual generation are acceptable but not particularly significant, considering the success of VAR. The approach appears to be a specific "two-scale" variant of VAR, replacing discrete tokenization (i.e., VQ) with continuous diffusion, inspired by MAR.
The authors are encouraged to provide deeper insights into the discussion and include necessary comparisons to "VAR w/ diffusion heads" to assess whether additional stages could lead to further improvements.
As I am not an expert in this area, I will seek input from other reviewers for a more objective evaluation. This recommendation is not final.

**Questions For Authors:**

N/A

**Relation To Broader Scientific Literature:**

N/A

**Theoretical Claims:**

N/A

---

> ### Author Rebuttal · Authors · 2025-04-01
>
> **Q1: Discussion on VAR and comparisons to "VAR w/ diffusion heads"**
>
> Thanks. We summarize the contributions of our Hi-MAR against VAR in two points:
>
> 1) VAR utilizes the low-resolution visual tokens directly to guide the next phase prediction, which would cause training-inference discrepancy as discussed in section 3.2. During training, VAR takes the ground-truth low-resolution tokens as the condition for the next phase. Since no ground-truth token is available at inference, VAR has to take the predicted noisy low-resolution tokens as the condition, resulting in training-inference discrepancy. Instead, to mitigate such discrepancy, Hi-MAR takes the conditional tokens estimated from the Hi-MAR Transformer of low-resolution visual tokens to trigger the second phase. As shown in Table 4, when replacing the pivots of visual tokens (similarly used in VAR) with our conditional tokens, the FID score clearly improves from 2.28 to 2.07, which validates the effect of the conditional tokens that mitigate training-inference discrepancy.
> 2) VAR models multi-scale probability distributions via a shared Transformer without additional guidance. This way leaves the inherent different peculiarities of each scale in autoregressive modeling not fully exploited, resulting in a sub-optimal solution for multi-scale token prediction. In contrast, our Hi-MAR incorporates a scale-aware Transformer block that provides scale guidance to the Transformer tailored to each phase.
>
> Moreover, as suggested, we experimented by implementing VAR with diffusion heads, and "VAR w/ diffusion heads" (FID: 2.67) manages to outperform VAR-d16 (FID: 3.30). Nevertheless, the performance of "VAR w/ diffusion heads" (FID: 2.67) is still inferior to our Hi-MAR (FID: 1.93), which demonstrates the effectiveness of hierarchical autoregressive modeling. We will add all discussions in revision.
>
> **Q2: Would further stages yield additional improvements**
>
> Appreciate this comment. We experimented by stacking one more stage (64x64 resolution) ahead of the low-resolution stage (128x128 resolution), and the FID score only fluctuates within the range of 0.03. We speculate that the low-resolution stage (128x128 resolution) has already provided sufficient global structure guidance for the next high-resolution stage. The use of an additional stage (64x64 resolution) might introduce unnecessary/redundant global structure information. We will discuss this in the revised version.
>
> **Q3: Autoregressive relationship of Figure 1 (a)**
>
> Thanks. To be clear, Figure 1(a) correctly illustrates the left-to-right autoregressive relations among the image token sequence. That is, each predicted token at position *i* (see the bottom output sequence) can only be emitted conditioned on the previous input tokens at positions less than *i* (see the top input sequence). We will clarify this in revision.

---

> > ### Comment · Reviewer_kLHw · 2025-04-03
> >
> > Thank you to the author for the clarification, which has resolved some of my concerns. I will maintain my rating as Borderline.
> >
> > Overall, I acknowledge the author's exploration in engineering and the empirical results. However, I remain concerned about the significance of the technical improvements of Hi-MAR compared to VAR. From the perspective of technical contribution, this is a fairly marginal paper (meaning it has a probability of being either accepted or rejected at any top ML/CV conference). Technically, it is a combination and repackaging of existing work, with incremental contributions, including:
> >
> > 1. Drawing inspiration from the multi-scale approach of VAR. Specifically, the authors explore a two-stage design.
> > 2. At each scale, the authors adopt a non-autoregressive masked prediction task, similar to the MaskGIT and MAGViT series. Compared to VAR’s “one-step prediction” approach at each scale, this can be seen as sacrificing some *computational efficiency* by using more inference steps in exchange for improved *prediction quality* within the scale.
> > 3. Additionally, to enhance the visual quality of the generated data, the authors replace the VQ operation with a diffusion head inspired by MAR, improving fidelity.
> >
> > #### Some further suggestions:
> > Conduct analysis experiments to evaluate the trade-off between effectiveness and efficiency. Specifically, starting from a two-stage VAR, progressively modify the approach by:
> > - Changing "directly predicting the next-scale feature map" to "iteratively predicting the next-scale feature map via masked prediction."
> > - Replacing the VQ head with a diffusion head.
> > Evaluate the impact of these changes on both performance and efficiency.
> >
> > Besides, Figure 4 in the main paper should include VAR for comparison.
> >
> > #### Question:
> > I understand the author’s mention of the "training-inference discrepancy in VAR"—I believe this is a common issue for most autoregressive models, namely, the accumulation of errors during inference. However, I am still unclear on why Hi-MAR alleviates this issue. The accumulation of errors should still occur in Hi-MAR’s inference process, whether during intra-scale multi-step masked prediction or next-scale prediction.

---

> > > ### Author Response · Authors · 2025-04-06
> > >
> > > ### Q1: Technical contribution
> > > Appreciate your response. We would like to provide a more detailed clarification on technical contribution of our Hi-MAR, especially compared to existing works like VAR:
> > > 1. **The use of conditional tokens to alleviate training-inference discrepancy across scales is novel**: We identify and alleviate the training-inference discrepancy across scales, which is a common yet underexplored issue in multi-scale prediction models (e.g., VAR, FlowAR, Muse). As shown in Table 4 of our paper, simply using low-resolution visual tokens as pivots to guide denser token prediction leads to marginal FID improvement (2.31 to 2.28), due to the inconsistency of pivot tokens between training and inference. To mitigate this, we propose using low-resolution **conditional tokens** generated by Hi-MAR Transformer to guide denser token prediction. This strategy ensures consistency between training and inference. As shown in Table 4, replacing the visual token pivots (as used in VAR) with our conditional tokens yields a notable FID improvement (2.28 to 2.07), which validates our proposal. A detailed explanation is provided in Q4 below.
> > > 2. **The proposal of scale-aware Transformer block is novel.** We introduce a scale-aware Transformer block that provides tailored scale guidance for each phase. This design is novel, effective, and not introduced in VAR. It is also worth noting that our hierarchical modeling can be easily applied to most VAEs without the need to train a multi-scale autoencoder, which needs to be trained in VAR.
> > >
> > > We therefore kindly invite Reviewer kLHw to reconsider assessment on our Hi-MAR's essential technical contributions, depending on the above discussions.
> > > ### Q2: Effectiveness-efficiency trade-off
> > > As suggested, we show a detailed comparison on effectiveness and efficiency across different methods in this new table (see the [link](https://anonymous.4open.science/r/HiMAR_FigTab/README.md)). Starting from a two-stage VAR, we progressively apply modifications: 1) Adopt masked autoregression for each stage (row 2 in this table); 2) Add a diffusion head for each scale (row 3 in this table). Note that we change the dimension and depth of VAR so that the parameter number of modified VAR is similar to Hi-MAR-B. As shown in this table, while these modifications improve performance, they still lag behind Hi-MAR in both accuracy and speed. Notably, even with these modifications, the best FID 2.30 of these variants only approaches that of Hi-MAR pivoting on ground-truth visual tokens (FID 2.28), whereas Hi-MAR further improves to 2.07 by introducing conditional tokens, highlighting the importance of addressing training-inference discrepancy.
> > > ### Q3: Figure 4 should include VAR
> > > Thanks. As suggested, we include VAR for comparison in the revised Figure 4 (see this [link](https://anonymous.4open.science/r/HiMAR_FigTab/README.md)). We will add this in revision.
> > > ### Q4: Training-inference discrepancy
> > > We clarify the discrepancy issue and how Hi-MAR resolves it. Let us consider a simplified two-scale setting.
> > > In VAR:
> > > - **Training**: The model learns to predict large-scale tokens $x_l$ conditioned on ground-truth small-scale tokens $x_s$, i.e., $P(x_l|x_s)$.
> > > - **Inference**: The model first predicts small-scale tokens $\hat{x}_s$, which may contain errors, and then uses them to predict $x_l$, i.e., $P(x_l|\hat{x}_s)$.
> > >
> > > This mismatch between $x_s$ (GT) in training and $\hat{x}_s$ (noisy) in inference introduces a training-inference discrepancy, leading to error accumulation and degraded generation quality.
> > >
> > > In Hi-MAR:
> > > - **Training**: In the first phase, a proportion of small-scale visual tokens $x_s$ are masked and the remaining unmasked ones $x_{s,v}$ are fed into Hi-MAR Transformer. The Hi-MAR Transformer outputs conditional tokens $z_{s,m}$, which are further fed into the diffusion head for predicting the masked tokens $x_{s,m}$ as MAR. In the second phase, similar masking procedure is also applied to the denser visual tokens $x_l$. Instead of using ground-truth $x_s$, Hi-MAR Transformer takes the small-scale conditional tokens $z_{s,m}$ from the first phase along with the unmasked visual tokens $x_{l,v}$ as input to generate denser conditional tokens $z_{l,m}$. Finally, the Diffusion Transformer head conditioned on $z_{l,m}$ is adopted to predict the denser masked tokens $x_{l,m}$.
> > > - **Inference**: We follow the same procedure (i.e., first predict small-scale conditional tokens $z_s$, and then predict denser conditional tokens $z_l$ based on $z_s$), ensuring the consistency of pivot tokens between training and inference.
> > >
> > > This design ensures that both training and inference in Phase 2 rely on predicted conditional tokens rather than ground-truth tokens. As shown in Table 4, this leads to a notable FID improvement (2.28 to 2.07), validating our proposal.

---

### Official Review · Reviewer_S3ag · 2025-03-17

**Overall Recommendation:** 3

**Summary:**

This paper improves Masked Autoregressive models (MAR) by introducing hierarchical modeling, specifically, low resolution is used as pivots. Additionally, MLP-based Diffusion is changed to global Diffusion to further improve the performance.

**Claims And Evidence:**

The claims made in this paper are supported by both qualitative and quantitative results.

**Essential References Not Discussed:**

None.

**Experimental Designs Or Analyses:**

Section 4.3 to 4.5 give a comprehensive and solid evaluation on the proposed method.

**Methods And Evaluation Criteria:**

The proposed method is evaluated on class-conditional image generation and text-to-image generation.

**Other Comments Or Suggestions:**

None.

**Other Strengths And Weaknesses:**

My biggest concern about this paper is the introduction of global Diffusion.

I admit it helps improve the overall performance, but it is well-known that an additional Diffusion module will improve the image generation ability no matter what methods are used before it.  The focus of this paper is supposed to prove the effectiveness of Hierarchical modeling, so my suggestion to Table 4 is to add a new row of "Pivots + MLP-based diffusion heads" to show the gain from Hierarchical modeling only.

**Questions For Authors:**

This paper provides an intuitive method to improve MAR and shows its effectiveness on a range of tasks, I tend to accept this paper.

**Relation To Broader Scientific Literature:**

This paper is closely related to autoregressive visual generation, which has broad impacts.

**Theoretical Claims:**

Not applicable.

---

> ### Author Rebuttal · Authors · 2025-04-01
>
> **Q1:  Add a new row of "Pivots + MLP-based diffusion heads" in Table 4**
>
> Appreciate this comment. As suggested, we conducted experiments by including a new ablated run of "Pivots + MLP-based diffusion heads" in Table 4. This ablated run enables hierarchical modeling with shared MLP-based diffusion heads, without using any additional Diffusion module. As shown in this table, the FID of this new ablated run (the third row) manages to outperform MAR (the first row) by 0.17, which clearly validates the effectiveness of the hierarchical modeling. We will add the discussion in revision.
>
> |       Pivots       |  Diff. Head_1 |  Diff. Head_2  | Scale vector |  #Para. | FID  |
> | ------------------ | ----------------------- | ------------------------ | ------------ | ------- | ---- |
> |        &cross;     |         &cross;         |         MLP-based        |    &cross;   |  208M   | 2.31 |
> |   visual tokens    |        MLP-based        |         MLP-based        |    &cross;   |  245M   | 2.28 |
> | conditional tokens |       &cross;   |          Shared MLP-based        |    &cross;   |  208M   | 2.14 |
> | conditional tokens |        MLP-based        |         MLP-based        |    &cross;   |  245M   | 2.07 |
> | conditional tokens |        MLP-based        |         Transformer      |    &cross;   |  239M   | 1.98 |
> | conditional tokens |        Transformer      |         Transformer      |    &cross;   |  233M   | 1.98 |
> | conditional tokens |        MLP-based        |         Transformer      |    &check;   |  242M   | 1.93 |

---

> > ### Comment · Reviewer_S3ag · 2025-04-09
> >
> > Thanks to the authors for providing additional experiments. After reading the rebuttal and other reviews, I keep the score.

---

### Decision · Program_Chairs · 2025-05-01

**Decision:**

Accept (poster)

**Comment:**

The authors present Hierarchical Masked Autoregressive models (Hi-MAR) for image generation. Specifically, Hi-MAR first generates low-resolution tokens, which are used to guide the generation of high-resolution tokens. The authors also propose Diffusion Transformer head to improve global context modeling. The effectiveness of the proposed method is demonstrated on ImageNet (class-condition) and COCO (text-to-image).

In the initial reviews, the reviewers expressed several concerns:

* Reviewer dCxF: Clarification of training-inference discrepancy issue; speed and accuracy behave for longer visual sequences; more Text-to-Image benchmark evaluations; inclusion of related works (Fluid, Infinity, FlowAR, HART, FlexVAR, FractalAR); ablation on the effect of phase 1 resolution, and so on.

* Reviewer SRgF: Limited novelty due to the incremental improvement over Muse, VAR, and HART; missing speed comparison with VAR and HART; ablation on different resolutions.

* Reviewer kLHw: Detailed comparison with VAR; ablation on additional stages; effectiveness-efficiency trade-off

* Reviewer S3ag: Additional ablation study for Tab. 4.

The provided rebuttal effectively assuaged the reviewers' concerns to some degree, while Reviewers kLHw and dCxF are still on Borderline. After carefully considering the reviews, author rebuttal, discussions, and the draft, the AC agrees with the reviewers that the proposed method is somewhat incremental, but appreciates the authors' efforts. As a result, the AC recommends acceptance of the paper.

However, the authors are strongly encouraged to incorporate feedback from the reviews and discussion into the final version. For example, it is critical to clearly discuss the difference between the proposed method and existing works (e.g., the discussion/comparison with VAR/Muse/HART/FlowAR). As a side note, the AC observes that the proposed scale-aware Transformer shares similarities with the scale-wise Flow Matching module in FlowAR; a more detailed discussion on this point would be greatly appreciated.